# Pre-Sowing Treatments, Seed Components and Water Imbibition Aids Seed Germination of *Gloriosa superba*

**Yogesh Ashok Mahajan** [1,2,3], **Balkrishna Ankush Shinde** [3], **Arun Torris** [4], **Akshay Baban Gade** [3], **Vipul Subhash Patil** [5], **C. K. John** [3], **Narendra Yeshwant Kadoo** [3,6,*] and **Tukaram Dayaram Nikam** [1,*]

[1] Department of Botany, Savitribai Phule Pune University, Pune 411007, India
[2] Horticulture Section, CSIR-National Chemical Laboratory, Pune 411008, India
[3] Biochemical Sciences Division, CSIR-National Chemical Laboratory, Pune 411008, India
[4] Polymer Science and Engineering Division, CSIR-National Chemical Laboratory, Pune 411008, India
[5] Catalysis and Inorganic Chemistry Division, CSIR-National Chemical Laboratory, Pune 411008, India
[6] Academy of Scientific and Innovative Research, Ghaziabad 201002, India
[*] Correspondence: ny.kadoo@ncl.res.in (N.Y.K.); tdnikam@unipune.ac.in (T.D.N.);
Tel.: +91-20-2590-2724 (N.Y.K.); +91-20-2560-1438 or +91-20-2560-1439 (T.D.N.)

**Abstract:** *Gloriosa superba* L. is a horticulturally and medicinally important plant. Its seeds have poor, erratic, and deferred germination. The detailed seed structure components and water imbibition mechanism facilitating the process of seed germination in *G. superba* remain unexplored. Therefore, it is essential to develop methods to ensure consistent and enhanced seed germination in *G. superba*. Various pre-sowing treatments along with the Brunauer–Emmett–Teller (BET) surface area analysis and 3D X-ray micro-tomography (micro-T) were employed to elucidate seed structure components, porosity network, and the water imbibition mechanism during germination in *G. superba*. The study revealed that consistent and significantly improved seed germination (>85%) was observed using the pre-sowing treatment mechanical scarification followed by 24 h water soaking in *G. superba*. BET and micro-T showed that the tegmen of the seed coat exhibited porosity (21%) with a well-connected porosity network (17.50%) that helped in water movement through hilum, which was confirmed by phosphotungstic acid staining. However, the sarcotesta and endosperm were water-impermeable due to their negligible porosity. Multidisciplinary techniques such as BET and micro-T along with conventional methodologies can be employed to address the seed coat structure, porosity, and water imbibition mechanism aiding seed germination. Mechanical scarification enabled the water to penetrate internal seed layers through the permeable tegmen via the reticulate pore network, which significantly improved seed germination. The developed seed germination method can produce a large number of plants in less time and conserve the natural populations of this high-value medicinally important species.

**Keywords:** micro-T; BET; seed components; seed dormancy; seed germination

## 1. Introduction

Germination of a dry seed starts with water imbibition through permeable seed coat layers, followed by the embryo growth. Environmental factors play a crucial role in seed drying during developmental stages toward maturity. When the moisture content in seeds of some plant species reach a threshold where seeds become dry, they can become impermeable [1–3]. In permeable seeds, raphe, micropyle, and hilum act as water entry points, while these features are closed in impermeable seeds. However, earlier examples in the literature show variable results for water entry in several plant species [4–7]. Although an impermeable seed coat in seeds is important for the long-term storage and survival of various plant species [8,9], it also strongly inhibits the imbibition and initiation of germination

[10,11]. Thus, the permeability of the outer seed coat plays a crucial role in seed germination [9]. The impermeable seed coat is generally observed in tropical plant species [2,12].

*G. superba* is a medicinally important perennial vine used as a commercial source of colchicine. It occurs across the tropics and is mainly found in Africa and Asia [13]. Due to its poor, erratic, and deferred seed germination, it is mainly propagated using tubers. However, its tubers have a very low multiplication rate with an annual life cycle producing a maximum of two tubers per plant in a year, which fails to match the demand [14–17]. Due to these reasons and high demand, the tubers are extracted from its wild habitats, which has threatened natural populations. Thus, there is an urgent need to develop methods to significantly improve seed germination in glory lily, so that their wild populations are conserved in its wild habitats.

Previously, the morphology of the ovule and seed structure of *G. superba* were studied with respect to seed production. The reports suggested that the removal of sarcotesta (outer seed coat) facilitated germination, but there was still poor and erratic germination, particularly at various temperature ranges; while the seeds with intact sarcotesta had extremely poor germination [18]. To overcome the impermeability of seed covering layer(s), pre-sowing treatments such as acid scarification, boiling/hot water treatment, and mechanical scarification have been employed to enhance seed germination [19]. Using 3D X-ray micro-tomography (micro-T) is an advanced technique for the three-dimensional visualization of internal plant structures [20]. Earlier studies employed micro-T for elucidation of cellular structures [21,22], internal seed structures [23,24], and the morphology of wheat grains [25]. Besides, Brunauer–Emmett–Teller (BET) surface area is an essential parameter that denotes the adsorption capacity of solid surfaces [26,27]. BET surface analysis was used to determine physical properties such as pore size, pore volume, and pore-volume distribution of coconut leaf and teak wood sawdust [28,29].

Hence, *G. superba*, despite being a valuable plant species, an effective pre-sowing method to obtain consistent and enhanced germination is still lacking. Therefore, the present study aimed to develop a pre-sowing treatment for significantly enhancing seed germination in *G. superba*. Although basic information about seed coat structures in *G. superba* is available, no attempts have been performed to analyze the porosity in seed coat layers and the water entry on imbibition resulting in germination. We have employed the BET analysis and 3D micro-T techniques to reveal the structure of seed coat layers, porosity, and water imbibition in seeds of *G. superba*.

## 2. Materials and Methods

### 2.1. Plant Material

Initially, the tubers of *G. superba* were collected from different parts of Western Ghats, Maharashtra, India [30]. The collected tubers were stored and planted in a plot at CSIR-National Chemical Laboratory, Pune, India. Standard agronomic practices were followed to raise healthy plants. *G. superba* pods were harvested on maturity and seeds were collected after drying the pods in sunlight for a fortnight. The harvested seeds were further dried in the shade for 5–6 weeks to reduce their moisture content and prevent microbial growth and physiological degradation. Similarly, the seeds were harvested and stored for three consecutive years (2018, 2019, and 2020). Furthermore, all the experiments were conducted in 2019 and 2020 to study germination efficiency.

### 2.2. Electrical Conductivity and Viability Test of G. superba Seeds

Three replicates of 100 seeds each from three different storage periods (1, 2, and 3 years after harvesting) were soaked in 50 mL sterile distilled water at 20 °C and electrical conductivity (EC) was measured after 24 h [31,32]. Similarly, the viability of three replicates of 100 seeds for each storage period was evaluated by treating the mechanically scarified seeds with 1% solution of tetrazolium chloride (TZ) [33,34]. After the treatment, the seeds were placed on sterile germination sheets for 24 h at 24 °C. After 24 h, the seeds were cut in two

equal parts to check their viability and the embryos turning pink or red on tetrazolium staining were considered viable, whereas those being colorless or black were deemed unviable.

### 2.3. Seed Surface Sterilization

The seeds were surface sterilized using 5% sodium hypochlorite (NaOCl) for 5 min and thoroughly rinsed four times with sterile water before applying any treatment.

### 2.4. Pre-Sowing Seed Treatments

Various seed pretreatments were evaluated for their effects on seed germination and seedling growth performance.

#### 2.4.1. Mechanical Scarification and Water Soaking

Mechanical scarification is the most commonly used scarification method. *G. superba* seeds were scarified manually with a sandpaper of grit size P60 and P100 until the seed's outer layer (sarcotesta) was removed. These scarified seeds were soaked in sterile water for different time durations (i.e., 0, 24, 48, 72, 96, and 120 h) and used in in vitro and in vivo germination experiments as mentioned before.

#### 2.4.2. GA$_3$ Treatment

One hundred *G. superba* seeds in each of three replicates were treated with various GA$_3$ concentrations (100 ppm, 200 ppm, and 300 ppm) for 60 min with intermittent stirring. The treated seeds were thoroughly rinsed four times with sterile water before sowing as mentioned before.

#### 2.4.3. Sulfuric Acid Treatment

The seeds were soaked in varying concentrations of sulfuric acid (H$_2$SO$_4$) (25%, 50%, and 75%) for 30 min. The treated seeds were thoroughly rinsed four times with sterile water before sowing as mentioned before.

### 2.5. Seed Germination Procedure

For in vitro trials, 100 seeds each in three replicates were placed on germination paper moistened with 5 mL of sterile water in Petri dishes. The Petri dishes were wrapped with plastic wrap and incubated at 24 °C in the dark. The Petri dishes were inspected every 48 h for germinating seeds and moistened by adding sterile water when required. The germinated seeds were removed and placed in seed germination trays filled with a sterile potting medium consisting of 3:1:1:1 (soil: manure: coco pit: sand) for further growth. The germination trays were placed in a glasshouse maintained at 24 °C during the day (12 h) and 20 °C during the night (12 h). The total number of germinated seeds was counted on the 30th day of incubation. Similarly, for in vivo experiments, 100 seeds each in three replicates were sown in germination trays filled with sterile potting medium. The germination trays were placed in a glasshouse maintained at 24 °C during the day (12 h) and 20 °C during the night (12 h). The trays were regularly watered with sterile water and the total number of germinated seeds was counted on the 30th day of sowing. Two trials were conducted, in vitro and in vivo.

### 2.6. Estimation of Abscisic Acid by LC-MS

The extraction and quantification of abscisic acid (ABA) from scarified seeds, non-scarified seeds, and sarcotesta were carried out using Liquid chromatography-mass spectrometry (LC-MS) as described by Perin et al. [35].

### 2.7. Seed Water Imbibition Capacity

The diameters of 50 randomly selected seeds of *G. superba* were measured using a Vernier Caliper. The combined dry weight of 50 seeds was recorded and the average dry weight of each seed was determined. The seeds were soaked in sterile water at room temperature for 0, 24, 48, 72, 96, and 120 h. The seeds were weighed (wet weight) and the diameters of the seeds were recorded at each time interval after imbibition. The changes in wet seed weight and diameter were evaluated [36].

### 2.8. BET Surface Area Analysis of Seeds by $N_2$-Adsorption-Desorption

About 100 mg crumbled seed samples (non-scarified seeds (NSS) and scarified seeds (SS)) were transferred to a dried quartz cell. The quartz cells were evacuated at 100 °C for 3 h in an inert gas to clean the surface by removing unwanted organic/inorganic molecules adhered to the solid surface. The cooled quartz cells were dipped in liquid nitrogen up to sample level and $N_2$ gas was passed with the flow rate of 20 mL/min for 6 h adsorption followed by 6 h desorption. The $N_2$ gas was used as a probe, which physically adsorbed and then desorbed on the solid surface. This formed mono- and multi-layers of the gas molecules on well-defined sites on the sample surface. The adsorption/desorption isotherm shows the nature of the porosity and surface properties of the examined samples. The Brunauer–Emmett–Teller (BET) surface area, total pore volume, and average pore size distribution were measured by $N_2$ sorption at −196 °C using the Autosorb IQ Instrument (Quantachrome Instruments, Boynton Beach, FL, USA). The surface area and pore size distribution of the seed samples were calculated using the BET equation and Barrett–Joyner–Halenda (BJH) method [37,38] as implemented in Quantachrome Autosorb software (Quantachrome Instruments, Boynton Beach, FL, USA).

### 2.9. 3D micro-T Imaging of G. Superba Seed Structure

*G. superba* seeds were mechanically scarified with sandpaper until the seed's outer layer (sarcotesta) was removed. The scarified, non-scarified, and water imbibed (i.e., 24 h and 48 h) seeds were imaged using micro-T (X radia 510 Versa X-ray Microscope, Zeiss, Pleasanton, CA, USA) to study their internal structure, morphology, porosity, and pore-size distribution. A seed was loaded onto the sample holder kept between the X-ray source and detector assembly. The detector assembly consisted of a scintillator, an objective lens, and a CCD camera. The X-ray source was ramped up to 60 kV and 5 W. The tomographic image acquisitions were completed by acquiring 3201 projections over 360° of rotation, with a pixel size of 3.7 microns (medium resolution), for every seed of approx. 4 × 4 mm size. Tomographic image acquisitions with a pixel size of 0.7 microns (high resolution) were also recorded. Besides, projections without the samples in the beam (reference images) were also collected and averaged. A phase-enhanced imaging protocol was adopted to increase the contrast of the resulting images. Scarified seeds were subjected to imbibition in 5% solution of phosphotungstic acid (PTA) for 24 h and 48 h to map hydration as PTA also is imbibed along with water. PTA-stained seeds were later dried in ambient air and imaged as described. The filtered back-projection algorithm was used to reconstruct the projections to generate two-dimensional (2D) virtual cross-sections (transverse sections) of the seeds. The image processing software Dragonfly Pro Version 3.6 [39] was used to generate volume-rendered three-dimensional (3D) images of seed samples. Segmentation and further processing were also performed using the same software. The 2D images were trimmed down to a sub-volume, filtered to remove noise, and segmented after manual threshold selection based on local minima from the grayscale histogram. The resultant 3D reconstructed volume was used to estimate the pore characteristics, where the pore radius was determined by fitting spheres into the pore volume.

### 2.10. Seedling Growth Performance

The growth performance of ten randomly selected seedlings of different treatments at 60 and 120 days after sowing was recorded. Various seedling growth parameters were recorded,

including shoot length (cm) and vigor index [40] at 60 days after sowing. Similarly, the lengths of the long arm and short arm of the tubers were measured at 120 days after sowing.

### 2.11. Statistical Analysis

The seed germination percentage data were normalized using arcsine transformation. The statistical analysis was carried out by Duncan's multiple range test (DMRT) with one-way ANOVA using SPSS 20.0 software (http://www.spss.co.in; accessed on 13 May 2021). The results were represented as mean ± SE (standard error of the mean) of three independent replicates for the respective experiments.

## 3. Results

### 3.1. Seed EC and Viability Test

Non-scarified seeds showed significantly less germination than scarified seeds ($p <$ 0.05). However, the viability of scarified seeds was >85% (Figure S1). The EC of seeds for different storage periods was assessed and compared with seed viability, which showed an increasing trend with increasing storage duration. However, the percent seed viability drastically declined after two years of storage (Figure 1).

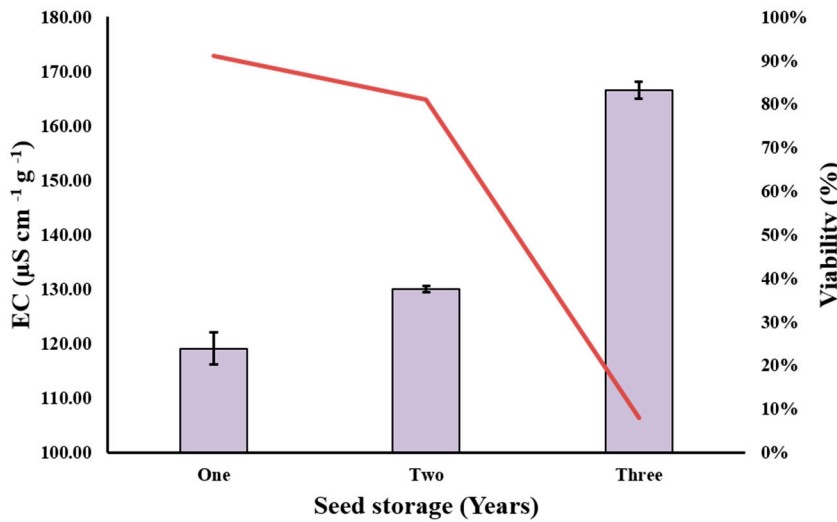

**Figure 1.** Effect of storage period on viability of seeds of *G. superba* using electrical conductivity (EC) test. The red line indicates the viability (%), while the bars indicate EC in µS cm⁻¹ g⁻¹.

### 3.2. Pre-Sowing Seed Treatments

Several pre-sowing treatments were evaluated to break the seed coat-imposed dormancy in *G. superba* and develop an effective method to enhance seed germination.

### 3.2.1. Mechanical Scarification and Water Soaking

Mechanical scarification (i.e., removing sarcotesta) followed by water soaking for various durations was carried out. Among all the treatments, a significantly higher germination percentage ($p < 0.05$) was observed in the "scarification followed by 24 h water soaking" treatment at room temperature than the control (non-scarified seeds). The germination percentage in the in vitro and in vivo experiments was 85% and 81%, respectively, compared to only 23% in the case of control seeds. The minimum number of days required for germination in the in vitro experiment was 5 and in vivo experiment was 11 days. However, no significant difference in germination percentage was noted for other treatments (Figure S2; Tables 1 and 2) irrespective of the duration of water soaking.

**Table 1.** Effect of physical and chemical treatments on seed germination of *G. superba* in in vivo condition 30 days after sowing (DAS) (*n* = 100).

| Treatment | Treatment Time (min/h) | Germination (%) 30 DAS (Mean ± SE) $ | Minimum Time Required for Germination (days) (days ± SE) $ |
|---|---|---|---|
| Control (without MS and WS) * | 0 h | 23 [g] ± 2.98 | 34 [h] ± 3 |
| MS (without WS) | 0 h | 56 [d] ± 1.50 | 17 [d] ± 2 |
| MS + WS | 24 h | 81 [a] ± 2.31 | 11 [a] ± 1 |
| MS + WS | 48 h | 79 [a] ± 1.30 | 13 [b] ± 1 |
| MS + WS | 72 h | 77 [b] ± 1.01 | 11 [a] ± 1 |
| MS + WS | 96 h | 70 [c] ± 1.52 | 11 [a] ± 1 |
| MS + WS | 120 h | 57 [d] ± 1.61 | 11 [a] ± 2 |
| GA$_3$ (100 ppm) | 60 min | 35 [e] ± 1.23 | 14 [c] ± 2 |
| GA$_3$ (200 ppm) | 60 min | 28 [f] ± 1.09 | 18 [d] ± 3 |
| GA$_3$ (300 ppm) | 60 min | 28 [f] ± 1.51 | 24 [f] ± 1 |
| H$_2$SO$_4$ (25%) | 30 min | 41 [e] ± 2.12 | 20 [e] ± 3 |
| H$_2$SO$_4$ (50%) | 30 min | 20 [g] ± 0.98 | 37 [i] ± 4 |
| H$_2$SO$_4$ (75%) | 30 min | 8 [h] ± 1.23 | 29 [g] ± 3 |

Notes: The values are means of 100 seeds. * MS—mechanical scarification; WS—water soaking; DAS—days after sowing [$]: the values appended with similar letters indicate non-significant differences, while those appended with different letters indicate significant differences (*p* < 0.05); SE: standard error of the mean.

**Table 2.** Effect of physical and chemical treatments on seed germination of *G. superba* in in vitro condition 30 days after sowing (DAS) (*n* = 100).

| Treatment | Treatment Time (min/h) | Germination (%) 30 DAS (Mean ± SE) $ | Minimum Time Required for Germination (days) (days ± SE) $ |
|---|---|---|---|
| Control (without MS and WS)* | 0 h | 33 [f] ± 1.56 | 37 [j] ± 1 |
| MS (without WS) | 0 h | 65 [c] ± 1.23 | 09 [d] ± 1 |
| MS + WS | 24 h | 85 [a] ± 3.21 | 06 [b] ± 1 |
| MS + WS | 48 h | 80 [a] ± 2.09 | 08 [c] ± 2 |
| MS + WS | 72 h | 73 [b] ± 1.81 | 05 [a] ± 1 |
| MS + WS | 96 h | 65 [c] ± 1.72 | 06 [b] ± 1 |
| MS + WS | 120 h | 59 [d] ± 1.31 | 06 [b] ± 2 |
| GA$_3$ (100 ppm) | 60 min | 65 [c] ± 1.23 | 11 [f] ± 2 |
| GA$_3$ (200 ppm) | 60 min | 70 [b] ± 1.29 | 07 [bc] ± 2 |
| GA$_3$ (300 ppm) | 60 min | 69 [b] ± 1.21 | 10 [e] ± 1 |
| H$_2$SO$_4$ (25%) | 30 min | 46 [e] ± 2.12 | 18 [g] ± 1 |
| H$_2$SO$_4$ (50%) | 30 min | 25 [g] ± 0.98 | 19 [h] ± 2 |
| H$_2$SO$_4$ (75%) | 30 min | 10 [h] ± 1.23 | 20 [i] ± 2 |

Notes: The values are means of 100 seeds. *MS—Mechanical scarification; WS—water soaking; DAS—days after sowing [$]: Numbers appended with similar letters indicate non-significant differences, while those appended with different letters indicate significant differences (*p* < 0.05); SE: Standard error of the mean.

### 3.2.2. GA$_3$ Treatment

The germination percentage was more or less similar irrespective of the GA$_3$ concentrations in the in vitro and in vivo conditions (Figure S3; Tables 1 and 2). In the in vivo conditions, the germination varied from 28% to 35%, while in the in vitro conditions, it ranged from 65% to 70%.

### 3.2.3. Sulfuric Acid Treatment

The seeds treated with dilute sulfuric acid exhibited higher germination than those treated with high concentration of sulfuric acid. The minimum (up to 8%) and maximum (up

to 46%) percentage of seed germination was recorded in 75% $H_2SO_4$ and 25% $H_2SO_4$ treatments, respectively, in either in vitro or in vivo experiments. The minimum time taken in days for radical emergence was also adversely affected by all the acid treatments (Figure S3; Tables 1 and 2).

### 3.3. ABA Quantification

The LC-MS analysis revealed negligible or no traces of ABA in all the studied seed samples (Figure S4).

### 3.4. Water Uptake Capacity of G. Superba Seeds

The rate of water uptake was higher during the first 24 h and later slowed down to reach the highest at 48 h. The rate of water uptake progressively declined after that. Similarly, the increment in seed diameter was maximum during 24 h to 48 h, while the minor variation in seed diameter was found until 120 h of water soaking (Figure 2; Table S1).

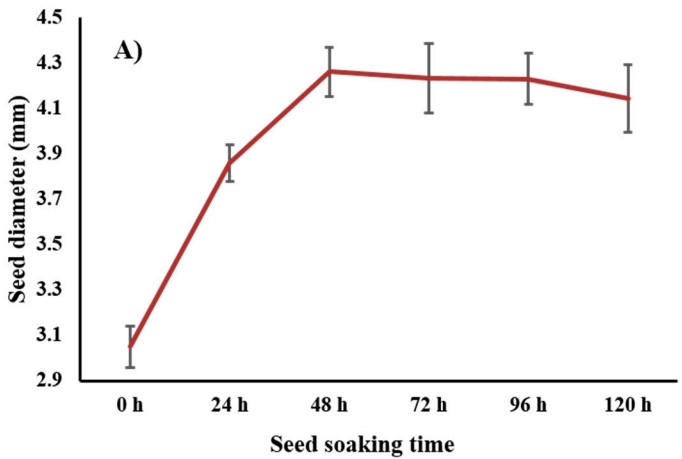

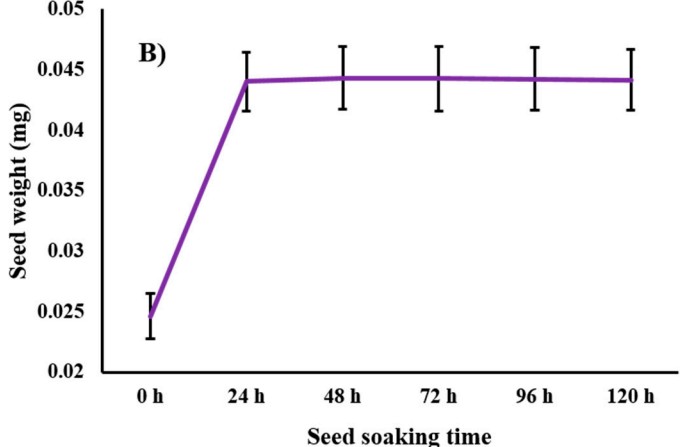

**Figure 2.** Determination of water uptake capacity measuring (**A**) diameter (mm) and (**B**) weight (mg) of seeds of *G. superba* soaked in water for different time intervals. Bar indicates standard deviation.

### 3.5. Porosity Analysis of G. Superba Seeds by BET

The BET analysis was performed using the $N_2$ adsorption–desorption isotherms at 196 °C according to the Barrett–Joyner–Halenda (BJH) method [37]. The shape of the adsorption isotherm for the scarified seed (SS) was type IV [29,38] and the hysteresis loop H2 type was observed. This showed that SS is associated with capillary condensation in the mesoporous. In the initial part of the isotherm, the relative pressure p/p° ≈ 0.05 of the SS sample can be

attributed to mono-multilayer adsorption. In contrast, the relative pressure p/p° was not found in the non-scarified seeds (NSS), which signifies that there were no pores on the surface. Hence, the surface area in NSS was not noticed due to negligible porosity, which was indicated by a reverse isotherm pattern for absorption and desorption compared with scarified seeds' porous surfaces (Figure 3A,B). The surface area (11.77 m²/g), total pore volume (0.0098 cc/g), and average pore volume (1.67 nm) were noticed in SS, whereas the same were not detected in NSS (Table S2).

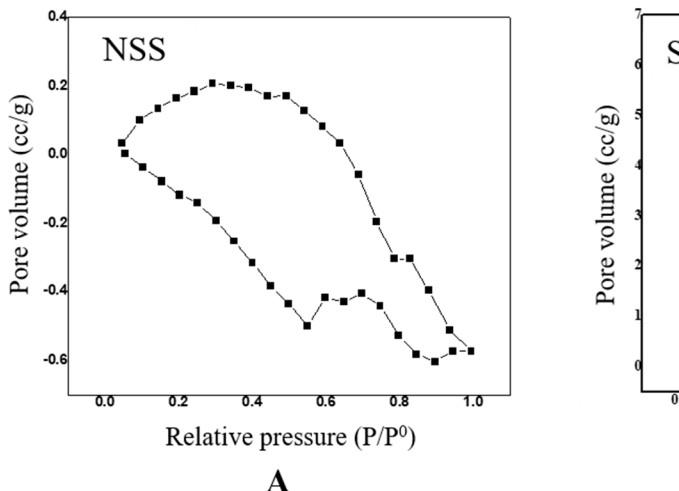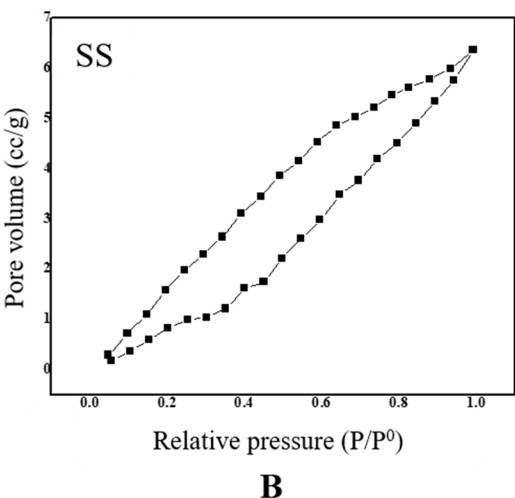

**Figure 3.** BET surface analysis (N2-adsorption–desorption) of *G. superba* seeds (**A**) Non scarified seed (NSS), (**B**) Scarified seed (SS).

*3.6. 3D micro-T imaging of G. superba Seed Structure*

The 2D and 3D images of NSS (Figure 4A1–A3) and SS (Figure 4B1–B3) were inspected to study the internal and external features of seeds in which sarcotesta, tegmen, and endosperm were observed (Figure 4; Video S1 (https://youtu.be/-59gs3GLBVE; accessed on 08 June 2021), and Video S2 (https://youtu.be/85Ae12Mu8LI; accessed on 8 June 2021). Furthermore, the phenotypic details such as seed length, width, thickness, and volume of SS at 0, 24, and 48 h were recorded (Table S3). These phenotypic parameters (length—4.4 mm, width—3.3 mm, thickness—2.9 mm, and volume—14.60 mm³) were found to be significantly increased ($p < 0.05$) at 48 h after imbibition (Figure S5). Based on these observations, a detailed study of all the seed structure was carried out. This study showed the detailed segmentation of sarcotesta, tegmen, and endosperm, which included virtual image stacking (Figure 5A) followed by demarcation at a higher resolution of the specific area (Figure 5B) of respective layer, segmented 3D image (Figure 5C), visualization of pores (Figure 5D) along with their color-coded pore volume distribution histogram (Figure 5E). In the seed structure, the pore volume was observed in a color-coded scale from purple to red, which corresponds to smaller to larger pores sizes. The sarcotesta displayed varied colored pores with the least pore volume. Similarly, endosperm also showed low pore volume, wherein purple color pores were predominantly found. In contrast, the tegmen exhibited higher pore volume distribution dominated by red, blue, and purple color codes (Figure 5; Video S3 (https://youtu.be/78xTpYN1s1o; accessed on 8 June 2021), Video S4 (https://youtu.be/TTGOBFxS97o; accessed on 8 June 2021), and Video S5 (https://youtu.be/Ga0RmYzBups; accessed on 8 June 2021).

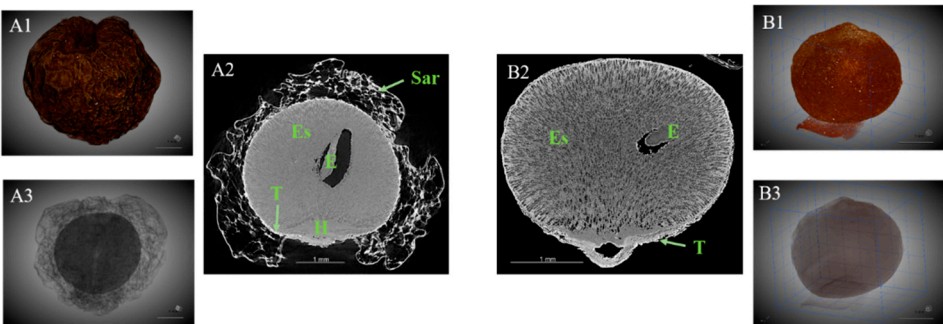

**Figure 4.** Illustration of seed structure of *G. superba* seed by micro-T. (**A1–A3**) Non-scarified seed; (**B1–B3**) Scarified seed; Sar—sarcotesta, T—tegmen, Es—endosperm, E—embryo, H—hilum.

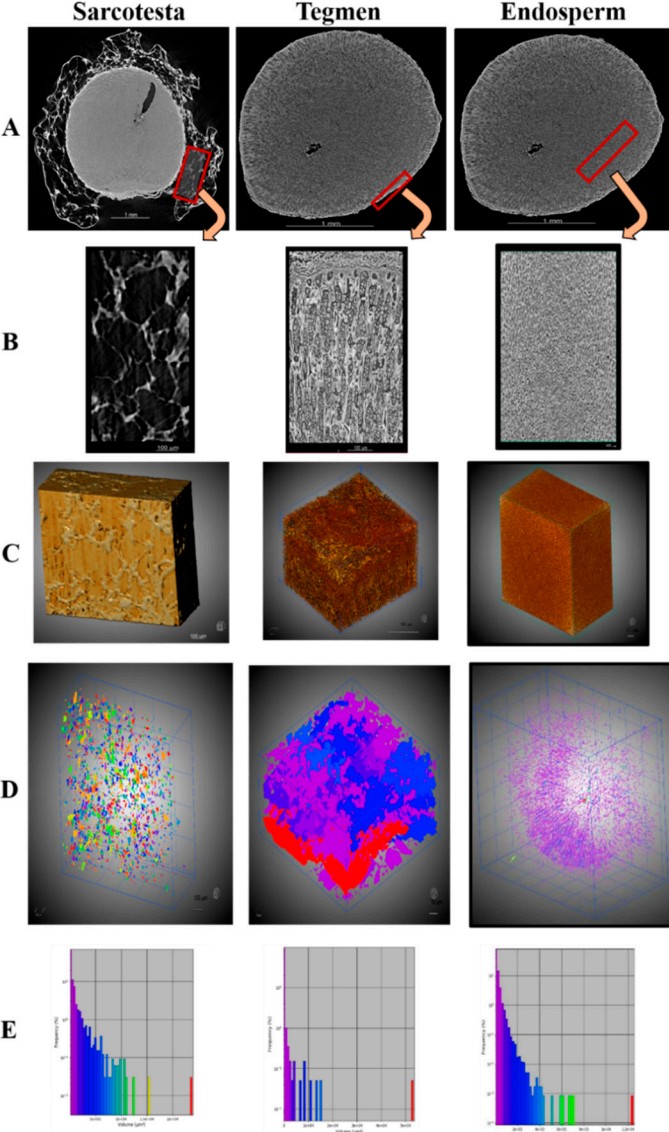

**Figure 5.** Detailed segmentation of pores in sarcotesta, tegmen, and endosperm; (**A**) Virtual image stack, (**B**) Demarcation, (**C**) Segmented 3D image, (**D**) Visualization of pores, (**E**) Color-coded pore volume distribution histogram; Colour scale- large (red) to small (purple).

The overall porosity and pore network in the seed structures were also quantified. The porosity in sarcotesta and endosperm was the lowest and the connectivity among the pores was also negligible. Additionally, the tegmen showed 21% overall porosity, of which 17.5%

was connected (Table S4). These observations of tegmen were also supported by 2D images of the pore network in tegmen of *G. superba* seed. These networks clearly showed connectivity among the pores and formed channels in tegmen (Figure S6; Video S6 (https://youtu.be/4AvZ-G_xZcA; accessed on 8 June 2021). To further evaluate water mobility in scarified seeds through the tegmen, the seeds also were soaked in 5% solution of phosphotungstic acid (PTA) for 0, 24, and 48 h to quantify the volume percentage of water imbibition (Video S7 (https://youtu.be/yYDayFDbbAs; accessed on 8 June 2021), Video S8 (https://youtu.be/JS0ka9FmDMY; accessed on 8 June 2021), and Video S9 (https://youtu.be/KLucSZ4PB2g; accessed on 8 June 2021). The volume percentage indicated by bright white staining was recorded at 11% at 24 h, which further increased to maximum saturation of 11% at 48 h (Figure 6; Video S8 (https://youtu.be/JS0ka9FmDMY; accessed on 8 June 2021) and Video S9 (https://youtu.be/KLucSZ4PB2g; accessed on 8 June 2021). In addition, the germinated seeds of *G. superba* were also visualized using micro-T (Figure S7; Video S10 (https://youtu.be/OVRcO5cFC98; accessed on 8 June 2021).

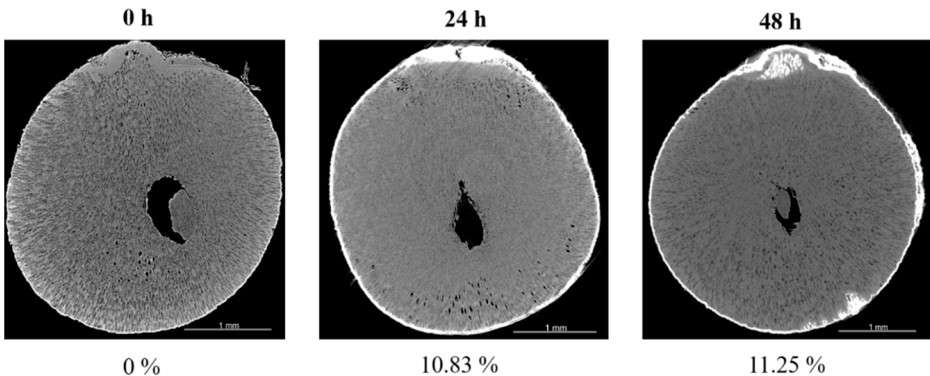

**Figure 6.** Mapping of hydration volume percentage inside dried mechanically scarified seeds of *G. superba* using micro-T and phosphotungstic acid (PTA) stain (at 0, 24, and 48 h of imbibition).

*3.7. Seedling Growth Performance of G. superba*

The shoot length of plant from "mechanically scarified and 96 h water-soaked seeds" was significantly higher than those of plants from other seed treatments ($p < 0.05$), including control. The tuber long arm length (on the 120th day of sowing) was considerably longer in scarified seeds followed by 24 h normal water soaking treatment compared to control. Furthermore, the tuber short arm length was also longer in plants from scarified and 24 h-soaked seeds than those from control seeds. The vigor index varied significantly among all the treatments ($p < 0.05$); however, it was higher in scarified and 48 h-soaked seeds. The scarification followed by all other water soaking treatments showed at par observations (Figure S8 and Table S5).

## 4. Discussion

*G. superba* is a medicinal plant used as a commercial source of colchicine. Although each plant produces several hundred seeds, only one or two tubers are produced. However, as *G. superba* seeds exhibit poor, erratic, and deferred seed germination, tubers are used as planting material [18,30,33]. This limits the planting material available to grow in the next season. Thus, bringing a large area under cultivation of the crop is a very slow process that can require several years. Hence, this study was conducted to develop a pre-sowing treatment to significantly enhance the seed germination in *G. superba*, so that a large area can be brought under its cultivation in a very short time. This would ensure regular supply of the raw material for commercial extraction of colchicine.

Before seed germination experiments, seed EC and viability tests were performed, which showed higher EC and lower viability percentage as the seed storage period increased. This might have resulted in seed deterioration caused by aging [31,32]. However, the tests clearly

indicated that *G. superba* seeds remain viable up to two years of seed storage at room temperature. Furthermore, various pre-sowing treatments such as mechanical scarification, mechanical scarification followed by water soaking for different time intervals, GA₃ treatment, and acid scarification for their effects on seed germination, were evaluated, along with seedling growth performance. Enhanced and consistent seed germination was observed after removing the nearly water-impermeable outer seed coat (sarcotesta) by mechanical scarification followed by water soaking for 24 h at constant temperature in in vitro conditions. In comparison, a previous study showed irregular germination percentage at varied temperature ranges [18]. However, the complete removal of the seed coat with no damage to the embryo is challenging [36]. Instead, removing only the sarcotesta with the least or no damage to tegmen, endosperm, and the embryo is a much simpler method resulting in better germination in *G. superba* seeds. It was also observed that mechanical scarification alone nearly doubled the seed germination percentage compared to the control in both in vivo and in vitro conditions. The treatment also significantly reduced ($p < 0.05$) the time required for radicle emergence.

GA₃ treatment is commonly used in different plant species to break physiological dormancy [41]. However, in the present study, the GA₃ treatment was found to be less effective as compared to mechanical scarification followed by water soaking. Similarly, the acid scarification treatment showed only a marginal difference in germination percentage compared to control, while the higher acid concentrations were detrimental. Furthermore, high levels of ABA in seeds have been reported in several plant species having physiological seed dormancy. The ABA levels were estimated in *G. superba* seeds; however, very low or no ABA traces were observed [42]. This may indicate the absence of physiological dormancy in *G. superba* seeds [11,43]. Thus, this type of dormancy can be attributed to the seed coat structure [11]; removing the seed coat is essential to overcome this seed dormancy. However, chemical and mechanical dormancy is currently considered physiological dormancy [44].

Mechanical scarification followed by 24 h water soaking yielded the highest germination percentage compared to all other treatments. The growth performance of the seedlings derived from scarified and water-soaked seeds was considerably greater than that of the control seedlings [36]. Overall, the mechanical scarification followed by 24 h water soaking was found to be the best pre-sowing treatment to enhance seed germination in *G. superba*. These results support previous studies, which reported that the dormancy in *G. superba*. is due to the least water-permeable seed coat (sarcotesta) acting as a physical barrier [11,18,33].

Seed germination starts with rapid water uptake, followed by seed swelling and further embryo enlargement [45]. A similar process of water uptake was witnessed in scarified and water-soaked seeds of *G. superba*. The scarified seeds imbibed water quickly, resulting in the rapid increase in seed weight and seed diameter until 48 h of soaking and, after this, the decline in seed diameter and weight was noticed, which seems to be its maximum saturation point [2,3,7]. Thus, the mechanically scarified seeds were more capable of absorbing water than non-scarified seeds. Similar reports in various plant species put forth that, initially, the water uptake was rapid, although it was found to be declined after threshold saturation [1,45,46].

Furthermore, to understand the porous nature of the seed coat layers and to gain better insight on water imbibition through the porous seed coat layers comprising the sarcotesta (outer layer), tegmen (second layer), and endosperm, we studied the surface area and porosity of scarified and non-scarified *G. superba* seeds using the BET method. Previously, BET was also used to measure the porosity and surface area of coconut leaves, its fiber, and sugarcane bagasse [29,47]. The BET surface area analysis showed that the scarified seeds were more porous with a larger surface area than the non-scarified seeds. This might have led to increased water imbibition capability of scarified seeds, which facilitated seed germination.

The outer nearly water-impermeable seed coat layer obstructs the entry of water and the exchange of gases [24]. However, very little information is available about the porous nature of seed coat structures and the mechanism of water imbibition in the case of *G. superba*. Using 3D micro-T can assist in measuring the internal structure of biological samples [48]. It has been reported that physically dormant seeds have thick, hard, and non-elastic testa [49,50].

Moreover, it is also known that water permeability and the hardening of testa vary among different plant families and species [51]. The micro-T analysis indicated that the second seed coat layer (tegmen) was significantly porous ($p < 0.05$) compared to sarcotesta and endosperm, which corroborates the observation that the tegmen was water-permeable [23,52,53]. We also observed a strongly connected pore network leading to water channels in the tegmen of *G. superba* seeds, whereas endosperm showed the least porosity and no connectivity among pores. The porosity network observed in tegmen was found to reach the hilum from all sides and further reached the embryo, resulting in the initiation of seed germination after water imbibition. Earlier findings also showed the hilum as an important structure for the entry of water in some species of Phaseoleae [54,55]. The difference between total pore volume and imbibed water volume might indicate air space required for gas exchange, which is also essential for seed germination [24]. Thus, micro-T helped to visualize and measure porosity along with connected water pore network in tegmen, which can enhance seed germination by a better understanding of the mechanism of water imbibition in *G. superba* seeds.

## 5. Conclusions

Glory lily is a horticulturally and medicinally important plant with a poor, erratic, and deferred seed germination. Currently, there are no simple and effective methods to improve seed germination for commercial cultivation. We found that *G. superba* seeds can remain viable up to two years after harvesting. Seed structure analysis using the BET surface area and micro-T techniques revealed the presence of nearly water-impermeable outer seed coat layer (sarcotesta), water permeable inner seed coat later (tegmen), and the endosperm. The techniques were used to visualize, quantify the porosity, and reveal the mechanism of water imbibition during the seed germination process. We observed a strongly connected pore network leading to water entry throughout the tegmen, which finally reached the embryo. Our study confirmed that the physical barrier imposed by the sarcotesta is responsible for the poor and erratic seed germination in *G. superba*. Among the various pre-sowing treatments, mechanical scarification followed by 24 h water soaking was found to be the best treatment. This resulted in consistent and enhanced (three-fold) seed germination percentage, reduced the number of days required for germination, and increased the seedling shoot length, vigor, etc. The above efficient, simple, reliable, and cost-effective pre-sowing treatment will help to generate a large number of *G. superba* plants in a short span of time. Thus, the dependency on *G. superba* tubers extracted from the wild will be much reduced or eliminated, conserving its natural populations and biodiversity. The micro-T and *BET* surface area analysis can also be explored for a better understanding of the seed structure and related mechanisms involved in seed germination of varied plant species.

**Supplementary Materials:** The following supporting information can be downloaded at: https://www.mdpi.com/article/10.3390/seeds2010002/s1, Figure S1: Effect of storage period on viability of seeds of *G. superba*: a. viable seeds are indicated by embryo stained pink and non-viable seeds are indicated by embryo stained black (after treating mechanically scarified seeds with 1% tetrazolium chloride (TZ) solution for 24 h); Figure S2: Germination of mechanically scarified seeds of *G. superba* in in vivo and in vitro condition after 60 days and in vitro condition after 30 days of sowing (n = 100 in each replicate), DAS—days after sowing, Control—without mechanical scarification (MS) and water soaking (WS) at 0, 24, 48, 72, 96, 120 h; Figure S3: Influence of GA$_3$ and H$_2$SO$_4$ treatments on *G. superba* seed germination in in vivo condition after 60 days and in in vitro condition after 30 days of sowing (n = 100 in each replicate), DAS—days after sowing, GA$_3$—treatment for 60 min and H$_2$SO$_4$ treatment for 30 min; Figure S4: LC-MS chromatograph for detection of abscisic acid in non-scarified and scarified seeds and sarcotesta of seeds of *G. superba*; Figure S5: Illustration of phenotypic details through micro-T of *G. superba* mechanically scarified seeds at 0, 24, and 48 h of imbibition; Figure S6: 2D image of pore network in tegmen of *G. superba* seeds by micro-T; Figure S7: Visualization of germinated seeds of *G. superba* using micro-T. (A) 2D cross-sectional image of germinated seed, (B) Volume-rendered 3D image of germinated seed, (C) Volume-rendered 3D image of germinated seed with defined grey-scale values; Figure S8: (A) *G. superba* seedlings after 60 days of germination, (B) Tuber of *G. superba* after 120 days of germination; Table S1: Effect of physical treatments on water uptake capacity of *G. superba* seeds during germination (n = 100); Table S2: BET surface analysis (N2-adsorption–desorption) of *G.*

*superba* seeds (n = 6); Table S3: Phenotypic details through micro-T of *G. superba* seeds (n = 6); Table S4: Porosity of *G. superba* seeds (n = 6); Table S5: Effect of physical and chemical treatments on growth performance of *G. superba* in in vivo condition 60 days after sowing (n = 100); Table S6: Porosity of *G. superba* seeds (n = 6); Table S7: Effect of physical and chemical treatments on growth performance of *G. superba* in in vivo condition 60 days after sowing (n = 100); Video S1: Non-scarified seed (NSS) structure of *G. superba* seed by micro-T Phase-3D (https://youtu.be/-59gs3GLBVE; accessed on 8 June 2021), Video S2: Scarified seed (SS) structure of *G. superba* seed by micro-T Phase-3D (https://youtu.be/85Ae12Mu8LI; accessed on 8 June 2021), Video S3: Detailed segmentation of pores in sarcotesta of *G. superba* seed by micro-T Phase 3D (https://youtu.be/78xTpYN1s1o; accessed on 8 June 2021), Video S4: Detailed segmentation of pores in sclerotesta of *G. superba* seed by micro-T Phase 3D (https://youtu.be/TTGOBFxS97o; accessed on 8 June 2021), Video S5: Detailed segmentation of pores in endotesta of *G. superba* seed by micro-T Phase 3D (https://youtu.be/Ga0RmYzBups; accessed on 8 June 2021), Video S6: Pore network in sclerotesta of *G. superba* seed by micro-T Phase 2D (https://youtu.be/4AvZ-G_xZcA; accessed on 8 June 2021), Video S7: Mapping of hydration volume percentage of phosphotungstic acid (PTA) stain inside *G. superba* non-scarified seed (NSS) by micro-T Phase 2D (https://youtu.be/yYDayFDbbAs; accessed on 8 June 2021), Video S8: Mapping of hydration volume percentage of phosphotungstic acid (PTA) stain inside *G. superba* mechanically scarified seed (SS) after 24 h of water imbibition by micro-T Phase 2D (https://youtu.be/JS0ka9FmDMY; accessed on 8 June 2021); Video S9: Mapping of hydration volume percentage of phosphotungstic acid (PTA) stain inside *G. superba* mechanically scarified seed (SS) after 48 h of water imbibition by micro-T Phase 2D (https://youtu.be/KLucSZ4PB2g; accessed on 8 June 2021); Video S10: Germinated *G. superba* seed using micro-T in Phase 2D and 3D (https://youtu.be/OVRcO5cFC98; accessed on 8 June 2021).

**Author Contributions:** Y.A.M., B.A.S., N.Y.K. and T.D.N. conceived and designed the experiments; Y.A.M., B.A.S. and A.T. performed micro-T experiments and generated the 2D/3D images and videos. Y.A.M. and V.S.P. performed BET experiments. Y.A.M., B.A.S. and A.B.G. performed all the other experiments and analyses. Y.A.M. and B.A.S. analyzed the data. Y.A.M., B.A.S., C.K.J., N.Y.K. and T.D.N. wrote, reviewed, edited, and finalized the manuscript. All authors have read and agreed to the published version of the manuscript.

**Funding:** This research and APC were funded by the Council of Scientific and Industrial Research (CSIR), India, Grant number: MLP101326.

**Institutional Review Board Statement:** Not Applicable.

**Informed Consent Statement:** Not Applicable.

**Data Availability Statement:** The authors declare that the data supporting the findings of this study are available within the paper and its supplementary information files. The videos depicting seed coat structures, porosity networks, and water imbibition are available on YouTube®.

**Acknowledgments:** All the authors are thankful to the gardeners of Horticulture Section, CSIR-National Chemical Laboratory, Pune, Tukaram Masal, Janardhan Waghmare, Rajappan Velluswami, and Jaishankar Ramachandran for their help during the seed germination experiments. Y.A.M. and B.A.S. are also thankful to Amey Bhide for his valuable inputs while drafting the manuscript. The authors thank T. Raja, Catalysis and Inorganic Chemistry Division, CSIR-National Chemical Laboratory, Pune, India, for providing access to the Brunauer–Emmett–Teller (BET) facility.

**Conflicts of Interest:** The authors declare no conflict of interest.

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
