# Peer review of "Pre-Sowing Treatments, Seed Components and Water Imbibition Aids Seed Germination of Gloriosa superba"

_2674-1024, doi:10.3390/seeds2010002_

Round 1
Reviewer 1 Report
Text is concern with seed water uptake in realation to seed coat structure and some other pre-sowing treatments.
Manuscript has some merits to get published.
There are number of suggestions that I suggest to get changed on the text.
You may take consideration them.

Author Response
Responses to Reviewers’ Comments
- A germination table is necessary to me
Response: As suggested, a germination table is included in the revised manuscript.
- Numbering the references?
Response: As suggested, the references are included in the manuscript.
- Extreme dryness may stimulate seed hardseededness? Any data what the storage conditions were?
Response: The seed drying method used is a standard method to dry the seeds. Moreover, the seeds of G. superba have fleshy outer seed coat (sarcotesta). Hence, the dryness method does not make the outer seed coat hard. Earlier publications also have mentioned about the erratic seed germination in G. superba. After drying, the seeds were stored at room temperature.
- Any maturation indicator for seeds? Colour, dryness etc.
Response: At maturity, the outer seed coat shrinks and the colour turns to red/dark red. Moreover, after shade drying for up to six weeks, the weight of seeds remains constant and does not reduce. This indicates that the seeds have been dried optimally.
- Made by hand or machine?
Response: Seeds were scarified manually by rubbing them on sandpaper of grit size P60 and P100. This has also been mentioned in the revised manuscript.
- Seed and solution rate?
Response: 100 seeds in 100 ml of GA3 solution (100, 200, and 300 ppm) per replicate was employed. This has also been mentioned in the revised manuscript.
- Seeds were dipped in [or] soaked?
Response: Seeds were soaked in respective treatments. This has also been mentioned in the revised manuscript.
- All seedlings or some selected and measured?
Response: Ten randomly selected seedlings were measured. This has also been mentioned in the revised manuscript.
- Why not observing EC and TTZ before the storage? Then how come we will see the difference during storage?
Response: After harvesting, the seeds of G. superba are generally dormant for a period of four to six months and hence are stored at room temperature for at least six months. Hence, we decided to conduct the EC and TTZ tests before the experimental trials. The experiment was further carried out to check the viability till 2- and 3-year storage period.
- You should describe that line shows TTZ values?
Response: The description is now added as per the suggestion.
- Please make it the columns thinner and remove the shaded lines behind the figure.
Response: The figure has been revised accordingly.
- percentages were fine please do not go further decimal ...
Response: The percent values have been revised in the manuscript.
- This is comment on GA plus part of material and methods You said already that in earlier on.
Response: The suggested changes are made in the revised manuscript.
- Changes in Fig 2
Response: The figure has been edited (A and B inserted in figure, x axes- time mentioned, representations of bars added) as suggested.
- Highlighted references??
Response: All the highlighted references are now added in the revised manuscript.
Reviewer 2 Report
Manuscript entitled "Pre-sowing treatments, seed components and water imbibition aids seed germination of Gloriosa superba" submitted to Seeds journal is well written and the results are presented in a logical and coherent manner. It is worth emphasizing the very good presentation and documentation of the research carried out. The paper is adequately organized and the topic is interesting and focuses on to develop a pre-sowing treatment for significantly enhancing seed germination in G. superba. The Autors employed the BET analysis and 3D micro-T techniques to reveal the structure of seed coat layers, porosity, and water imbibition in seeds of G. superba. This may contribute to facilitating the management of the seeds of this horticulturally and medicinally important plant.
The manuscript may be published in its current form in Seeds journal.
Author Response
Responses to Reviewers’ Comments
We would like to thank Reviewer for reviewing our manuscript and providing valuable observations and appreciation.
Reviewer 3 Report
This article study the seed germination of Gloriosa superba by several pre-treatment such as mechanical scarification and water soaking, GA3 treatment and sulfuric acid treatment. A conclusion is got that the mechanical scarification followed by 24 h water soaking is the best pre-sowing treatment method for improving the germination and its mechanism is provided by BET surface area analysis and 3D X-ray micro-tomography (micro-T). The work can supply great help in conserving natural population of Gloriosa superba. The article should be accepted after major revision.
The major concerns are as follow:
Line 2, porosity of seed components and water imbibition aids seed germination?
Line 16, 19, 22, 24, Gloriosa superba, need to be italic?
Line 164, 2.12.3. or 2.12 ?
Line 105-107, how to deal with “2.4. Pre-sowing seed treatments Various seed pretreatments were evaluated for their effects on seed germination and seedling growth performance.”
Line 202-204, “G. superba seed viability was tested using tetrazolium staining. The embryos turning pink or red on tetrazolium staining were considered viable, whereas those being colorless or black were deemed unviable.” Belong to Methods?
Line 206-207, “The EC for different storage periods was assessed and compared with seed viability” Belong to Methods?
Line 207, “The EC showed a significantly increasing trend”, unclear sentence.
Line 212-215, to delete “3.2. Pre-sowing seed treatments Several pre-sowing treatments were evaluated to break the physical or seed coat-im-213 posed dormancy in G. superba and develop an effective method to enhance seed germina-214 tion.”????
Line 217-218, to delete “3.2. Pre-sowing seed treatments Several pre-sowing treatments were evaluated to break the physical or seed coat-imposed dormancy in G. superba and develop an effective method to enhance seed germination.”??
Line 210-211, Title of Figure 1 is not complete and clear, for instance, what’s uscm-1g-1?
Line 227-228, to delete “GA3 is a well-known plant hormone routinely used for accelerating seed germination in various plant species.” Or put them into Introduction.
Line 241-242, to delet “ABA is a well-known physiological dormancy inducer in various plant species. The quantification of ABA in non-scarified, scarified seeds and sarcotesta was carried out” or put them in Methods?
Line 271, to delete “The seed structure of G. superba was visualized using the micro-T.” or put into Methods.
Line 293-294, “Illustration of seed structure of G. superba seed by micro-T. A1-A3: Non-scarified seed B1-B3) Scarified seed; Sar- sarcotesta, T-tegmen, Es- endosperm, E-embryo, H-hilum.”, unclear sentence, A1-A3)? :? Non-scarified seeds, B1-B3: Scarified seed. (in addition, title of Fig.3, Fig.5,need commas)
Line 296-298, to pay attention to punctuation symbol, Figure 5. Detailed segmentation of pores in sarcotesta, tegmen, and endosperm. A) Virtual image stack, B) Demarcation, C) Segmented, 3D image, D) Visualization of pores, E) Color-coded pore volume, distribution histogram, Colour scale- large (red) to small (purple) ???
Line 318, “The shoot length of Gloriosa seedlings was measured on the 60th day of sowing.” Put in Methods?
Author Response
Responses to Reviewers’ Comments
- Line 2, porosity of seed components and water imbibition aids seed germination?
Response: The sentence has now been modified to make it clearer to understand.
Old sentence - The porosity in seed components and water imbibition mechanism facilitating the process of seed germination remain unexplored.
New sentence - The detailed seed structure components and water imbibition mechanism facilitating the process of seed germination remain unexplored.
- Line 16, 19, 22, 24, Gloriosa superba, need to be italic?
Response: Gloriosa superba has been changed to italics at respective places.
- Line 164, 2.12.3. or 2.12 ?
- Line 105-107, how to deal with “2.4. Pre-sowing seed treatments Various seed pretreatments were evaluated for their effects on seed germination and seedling growth performance.”
Response: The error has now been corrected in revised manuscript.
- Line 202-204, “G. superba seed viability was tested using tetrazolium staining. The embryos turning pink or red on tetrazolium staining were considered viable, whereas those being colorless or black were deemed unviable.” Belong to Methods?
Response: The sentence has now been moved to the Materials and Methods section.
- Line 206-207, “The EC for different storage periods was assessed and compared with seed viability” Belong to Methods?
Response: The sentence has now been modified and merged with the next sentence, to make it easier to understand.
- Line 207, “The EC showed a significantly increasing trend”, unclear sentence.
Response: The sentence has now been merged with the previous sentence.
- Line 212-215, to delete “3.2. Pre-sowing seed treatments Several pre-sowing treatments were evaluated to break the physical or seed coat-imposed dormancy in G. superba and develop an effective method to enhance seed germination.”????
Response: We would like to keep the section, as we feel that it is relevant at the place and connects to the next sub-sections. The section numbers have been changed at respective places in the revised manuscript.
- Line 210-211, Title of Figure 1 is not complete and clear, for instance, what’s uscm-1g-1?
Response: The Fig 1, as well as, its description has been modified in revised manuscript to make it clearer.
- Line 227-228, to delete “GA3 is a well-known plant hormone routinely used for accelerating seed germination in various plant species.” Or put them into Introduction.
Response: The above sentence is now deleted from the revised manuscript.
- Line 241-242, to delete “ABA is a well-known physiological dormancy inducer in various plant species. The quantification of ABA in non-scarified, scarified seeds and sarcotesta was carried out” or put them in Methods?
Response: The above sentence is now deleted from the revised manuscript, as it is already mentioned in the Material and Methods section.
- Line 271, to delete “The seed structure of G. superba was visualized using the micro-T.” or put into Methods.
Response: The above sentence is now deleted from the revised manuscript, as it is already mentioned in the Material and Methods section.
- Line 293-294, “Illustration of seed structure of G. superba seed by micro-T. A1-A3: Non-scarified seed B1-B3) Scarified seed; Sar- sarcotesta, T-tegmen, Es- endosperm, E-embryo, H-hilum.”, unclear sentence, A1-A3)? :? Non-scarified seeds, B1-B3: Scarified seed. (in addition, title of Fig.3, Fig.5, need commas)
Response: The changes have been made at respective places in the revised manuscript.
- Line 296-298, to pay attention to punctuation symbol, Figure 5. Detailed segmentation of pores in sarcotesta, tegmen, and endosperm. A) Virtual image stack, B) Demarcation, C) Segmented, 3D image, D) Visualization of pores, E) Color-coded pore volume, distribution histogram, Colour scale- large (red) to small (purple) ???
Response: The changes have been made at respective places in the revised manuscript.
- Line 318, “The shoot length of Gloriosa seedlings was measured on the 60th day of sowing.” Put in Methods?
Response: The above sentence is deleted in the revised manuscript as it is already mentioned in the Material and Methods section.